# In Vivo Induction of Leukemia-Specific Adaptive and Innate Immune Cells by Treatment of AML-Diseased Rats and Therapy-Refractory AML Patients with Blast Modulating Response Modifiers

**DOI:** 10.3390/ijms252413469

**Published:** 2024-12-16

**Authors:** Michael Atzler, Tobias Baudrexler, Daniel Christoph Amberger, Nicole Rogers, Alexander Rabe, Joerg Schmohl, Ruixiao Wang, Andreas Rank, Olga Schutti, Klaus Hirschbühl, Marit Inngjerdingen, Diana Deen, Britta Eiz-Vesper, Christoph Schmid, Helga Maria Schmetzer

**Affiliations:** 1Medical Department III, Munich University Hospital, 81377 Munich, Germanydaniel.amberger@gmx.at (D.C.A.); alexanderrabe@hotmail.com (A.R.); ddeen@web.de (D.D.);; 2Bavarian Cancer Research Center (BZKF), 80539 Munich, Germanyklaus.hirschbuehl@uk-augsburg.de (K.H.); christoph.schmid@uk-augsburg.de (C.S.); 3Diaconia Hospital Stuttgart, 70176 Stuttgart, Germany; joerg.schmohl@diak-stuttgart.de; 4Department of Hematology and Oncology, Section for Stem Cell Transplantation, Augsburg University Hospital and Medical Faculty, 86156 Augsburg, Germany; 5Department of Immunology, Oslo University, 0027 Oslo, Norway; mariti@medisin.uio.no; 6Institute for Transfusion Medicine, Hannover Medical School, 30625 Hannover, Germany

**Keywords:** AML, new AML drugs, AML immunotherapy, leukemia-derived dendritic cells, kits, immune-monitoring, leukemia-specific cells

## Abstract

There is a high medical need to develop new strategies for the treatment of patients with acute myeloid leukemia (AML) refractory to conventional therapy. In vitro, the combinations of the blast-modulatory response modifiers GM-CSF + Prostaglandin E1, (summarized as Kit M) have been shown to convert myeloid leukemic blasts into antigen-presenting dendritic cells of leukemic origin (DC_leu_) that were able to (re-)activate the innate and adaptive immune system, direct it specifically against leukemic blasts, and induce memory cells. This study aimed to investigate the immune modulatory capacity and antileukemic efficacy of Kit M in vivo. Brown Norway rats suffering from AML were treated with Kit M (twofold application). Blasts and immune cells were monitored in peripheral blood (PB) and spleen. Upon the observation of promising immune modulatory effects in the treated animals, two patients with AML refractory to multiple lines of therapy were offered treatment with Kit M on an individualized basis. Safety, as well as immunological and clinical effects, were monitored. Samples obtained from a third patient in similar clinical conditions not receiving Kit M were used as controls for immune monitoring tests. *Animal experiments*: Drugs were well tolerated by the treated animals. After 9 days of treatment, DC_leu_ and memory-like T cells increased in the peripheral blood, whereas regulatory T cells, especially blasts, decreased in treated as compared to untreated control animals. *Clinical courses*: No severe side effects were observed. In patient 1482, PB blasts remained under the detection threshold during 27 days of treatment, thrombocytes were normalized, and (leukemia specific) immune effector cells of the adaptive and innate immune system increased up to 800-fold compared to the start of treatment. Patient 1601 responded with a 12% reduction in blasts in PB immediately after Kit M treatment. Several subtypes of (leukemia-specific) immune effector cells in PB increased up to four-fold during the 19 days of treatment. In contrast, immune-reactive cells decreased under mild chemotherapy in the PB of control patient 1511 with comparably refractory AML. Within the limitation of low numbers in both animal experiments and clinical applications, our data suggest that Kit M treatment of AML-diseased rats and patients is feasible and may induce leukemia-specific immune reactions and clinical improvement. A larger series and a prospective clinical trial will be required to confirm our observations. Beyond optimized doses and schedules of the applied compounds, the combination with other antileukemic strategies or the application of Kit M in less proliferative stages of the myeloid diseases need to be discussed. If effects are confirmed, the concept may add to the armamentarium of treatments for highly aggressive blood cancer.

## 1. Introduction

### 1.1. Acute Myeloid Leukemia

Acute myeloid leukemia (AML) is a high-risk disease. While patients with favorable genetics have a reasonable chance to be cured, the prognosis of AML classified as high risk based on genetic aberrations or insufficient response to induction therapy remains poor. Intensive induction chemotherapy followed by allogeneic hematopoietic stem cell transplantation (HSCT) is the only potentially curative treatment and represents the standard approach for these patients, especially at a younger age and with few comorbidities [1,2,3,4] For patients not eligible for high-dose chemotherapy, induction therapy using low-dose cytarabine or hypomethylating agents in combination with venetoclax [5] has considerably improved the results but without curative potential in the vast majority of patients. Beyond this, treatment options are limited, and the prognosis is grim for most patients with relapsed or refractory disease [6] Hence, there is a great need for new treatments. Among others, recent immune therapeutic approaches have addressed the dysfunctional reactivity of the immune system against leukemic blasts [7,8]

### 1.2. Dendritic Cell (DC)-Based Immunotherapy

DCs play a central role in connecting the innate and the adaptive immune system [9,10]. In vitro, DCs can be generated from CD14^+^ monocytes, which can be pulsed with leukemia-associated antigens (LAAs), leukemic peptides, or messenger RNA (mRNA, electroporation). Similarly, DCs can be generated from leukemic blast by culture techniques using combinations of granulocyte–macrophage colony-stimulating factor (GM-CSF) and a second response modifier (e.g., Picibanil, Prostaglandin E_1_ or E_2_). These compounds have been shown to be highly efficient DC-generating factors by providing danger signaling, enhancing DCs’ maturation and migratory capacity-without inducing blasts’ proliferation [11,12,13,14,15]. Those leukemia-derived DCs (DC_leu_) are characterized by the expression of the whole individual leukemic antigen repertoire, including leukemic antigens [11,16,17]. DC_leu_ can be re-administrated to patients as a kind of vaccine [18]. Alternatively, DC_leu_ can be induced in vivo after the systemic application of DC_leu_-inducing kits. As shown in vitro, the mode of action of GM-CSF in vivo is the induction of myeloid/dendritic cell differentiation; PGE1 increases the expression of CD197 and enhances dendritic cell maturation/migration. Hence, these kits may trigger DC/DC_leu_ differentiation and maturation [15,19].

The aim of the present study was to explore in vivo blast reprogramming/DC_leu_-inducing, as well as the tolerability, immune modulation, and antileukemic effects of Kit M (comprising GM-CSF and PGE_1_). Therefore, leukemia-diseased Brown Norway rats were systemically treated with Kit M, followed by clinical observation and immune monitoring. Upon promising results from the animal model, two voluntary well-informed patients with end-stage AML, who had given written informed consent, were treated by intravenous application of Kit M in individual salvage attempts.

## 2. Results

### 2.1. Animals

#### 2.1.1. Ex Vivo Priming of Rat T Cells with Kit M-Treated (DC/DC_leu_-Containing) Rat WB Results in Improved Antileukemic Reactivity

Ex vivo Kit M (vs. no Kit M) was shown to generate DC/leukemia-derived DCs from leukemia-diseased rats’ blood, leading to antileukemic immune activation after mixed lymphocyte culture (MLC, enriched with rats’ T cells) and to improved blast lysis (as demonstrated by chrome release assay. These data prove the ex vivo efficacy of Kit M-mediated immune activation/antileukemic efficacy in the animal model, as shown before with human blood [11,20].

#### 2.1.2. In Vivo Treatment of Healthy Rats with PGE1 Is Safe

To identify possible adverse effects, PGE1 was injected into three animals from the MHC identical healthy rat strain PVG.1N in comparable doses, as used in BNML rats (1 μg PGE_1_ per rat and injection). Rats were observed for 72 h for side effects such as interactions with other rats, skin irritations, altered breathing, sleeping behavior, weight, or mobility. No changes in physiological functions and behavior were found, suggesting a good tolerability and safety of the drug.

#### 2.1.3. In Vivo Treatment of BNML Rats with Kit M Is Safe and Leads to Antileukemic Immune Reactions

No adverse events were found in BNML rats under Kit M treatment compared to control animals. Compositions of immune-reactive cells in PB and spleens of Kit-treated vs. untreated rats were monitored after the sacrifice of rats. Cell subsets were analyzed as described in Appendix A. The application of Kit M to leukemia-diseased BNML rats increased DC_leu_ counts in the blood and spleen. Blast proliferation was not induced after treatment *(*Figure 1). Moreover, in the spleen and blood, a significant reduction in blasts compared to the control was seen after in vivo Kit M treatment (%blasts: PB: 10.7% (*p* < 0.0001); spleen: 15.8% (*p* < 0.033)). The frequencies of T_reg_ in blood decreased under the influence of Kit M. A trend of significant increase was shown for T_mem_ (memory-like CD8+ CD62L++ T cells) in PB after Kit M treatment (Figure 1). Frequencies of NK and NKT cells did not change throughout Kit treatment.

In summary, MLC demonstrated improved antileukemic reactions of DC_leu_-stimulated rat T cells ex vivo. Moreover, the treatment of leukemia-diseased rats with Kit M led to increased frequencies of DC/DC_leu_ and T_mem_ in vivo and, furthermore, to a significant reduction in blasts in rat PB and/or spleen.

Each of the three leukemia-diseased rats per group were treated with Kit M for 9 days (repeated injection of Kit M after 5 days) and then sacrificed. The composition of cells in PB and the spleen was quantified by flow cytometry. Data were statistically analyzed using a paired *t* test. Given are the mean values ± standard deviation, and differences were defined as significant with *p* values < 0.5. Abbreviations for cell subtypes are given in Appendix A.

**Figure 1 ijms-25-13469-f001:**
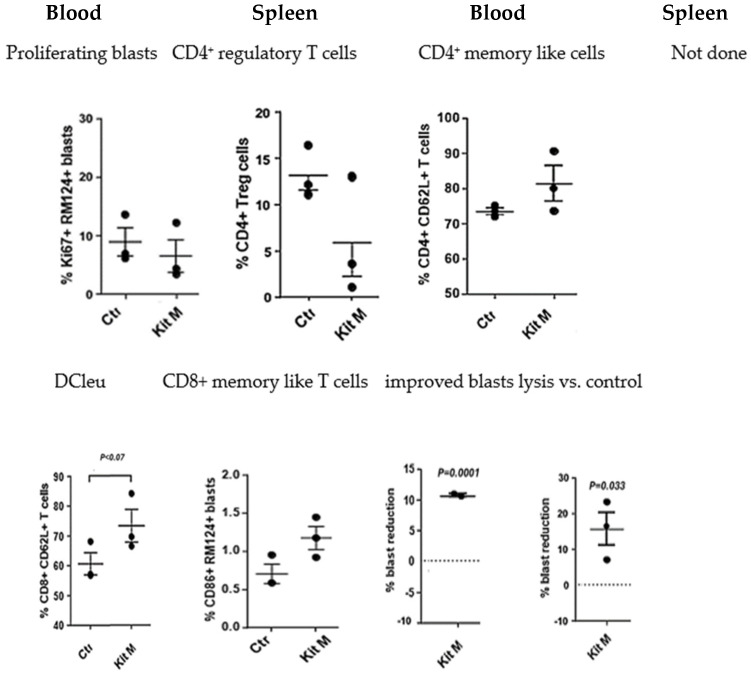
Treatment of leukemia-diseased rats with Kit M induced DC/DC_leu_ and DC/DC_leu_-activated immune-reactive and memory-like T cells and reduced regulatory T cells and blasts, preferentially in PB/spleen. Cell subtypes are given in Appendix A.

### 2.2. Patients

#### 2.2.1. Before Clinical Treatment: Successful Ex Vivo Generation of DC/DC_leu_ from Patients’ Blasts with Kit M

As described before [21], Kit M was shown to give rise to mature DC/DC_leu_ in all three patients’ blood samples ex vivo, demonstrated by the co-expression of proliferation markers on blasts. The induction of blast proliferation was not seen. After T cells enriched the MLC of immunoreactive cells with Kit M-pretreated DC/DC_leu_, we observed the activation of T cells. This came along with the induction of proliferating T_non-naive CD4+_/T_CD4+_ as well as of memory T cells (T_em/eff_/CD3+ or T_cm_/CD3+; cell subtypes are given in Appendix A), leading to improved blast lysis in Kit M-pretreated cells compared to controls.This observation provided the rationale to offer Kit M treatment in these particular patients.

#### 2.2.2. Clinical Courses and Immune Monitoring

Clinical characteristics of the two patients with refractory AML treated with Kit M and the control patient are provided in Table 1. Individual clinical and immunological courses under therapy are described here and summarized in Figure 2, Figure 3, Figure 4 and Figure 5.

#### 2.2.3. Patient 1482: Clinical Response, Improvement in Blood Counts, and Immunological Effects

Shortly before the start of blast-modulatory treatment with Kit M, the patient had Decitabine (20 mg/m^2^), followed by Hydroxyurea and low-dose Cytarabine (100 mg/m^2^), to minimize blast counts in PB (nearly 90%) and BM (nearly 70%). In the absence of any established therapy, he was offered the individual systemic salvage treatment as described here. Treatment was carried out on an inpatient basis over 4 weeks. Treatment started with the application of single drugs at low concentrations with dose escalation during treatment (see Appendix A for details).

Routine clinical and laboratory parameters showed that the treatment was safe and well tolerated, whereas the patient’s overall clinical conditions improved. An asymptomatic and transient decrease in blood pressure, known as a possible effect of PGE_1_, was seen once on the first day of application, representing the only CTC II° toxicity. It was completely resolved and did not occur again after prolongation of the infusion to >2 h. No other adverse events were seen during treatment.

Neutrophils in WBC increased from 10% to 50% and thrombocytes reached 100 G/l after 24 days (no need for platelet transfusions), whereas the total WBC and hemoglobin counts remained low. Treatment was stopped after 4 weeks, and the patient was discharged in good clinical condition. Unfortunately, 8 days later, progression of AML was seen with high blast counts in PB (40%) and BM (73%). The patient developed severe sepsis and died (Figure 2).

**Figure 2 ijms-25-13469-f002:**
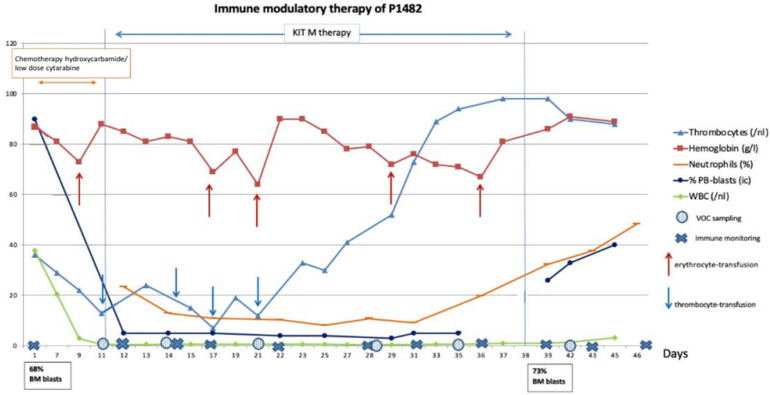
**Clinical course of disease of P1482 following low-dose chemotherapy and Kit M treatment.** Chemotherapy (hydroxycarbamide, cytarabine) was given from the start of observation to day 11. Kit M treatment was between day 11 and 38; no treatment was from day 38 to the end of observation. Blood cells (thrombocytes, hemoglobin, neutrophils, blasts) in peripheral blood (PB) and frequencies of BM blasts and leukocytes/white blood cells (WBCs) are given. ↑ Timepoints of erythrocyte transfusions; ↓ timepoints of thrombocyte transfusions. Immune monitoring at defined timepoints showed, in contrast to the samples obtained before treatment and from the patient without Kit M treatment, a continuous increase in DCs and proliferating CD8^+^ T cells, of T_non-naive_ (and a decrease in T_naive_) of the CD8 and CD4-lines, of Th_1_+ and Th_17_+, CD4^+^ T cells, and of Bcell_memory_ over the 4-week Kit M treatment. The same was true for T_cm_ and T_em_ of the CD8 and CD4-lines and for frequencies of NK cells (either CD161^+^ or CD56^+^), of CIK cells (either CD161^+^ or CD56^+^), Can be removed and iNKT cells (of the NK- as well as the CD3-type) (Figure 5A, P1482).

Antigen-specific cells were monitored after LAA stimulation by CSA. The overall (slightly) increasing frequencies of IFN-γ-producing CD4, CD8, CIK, and iNKT cells of the innate immune system were seen suggesting an in vivo production/activation of (potentially leukemia-specific) cells. Immune stimulatory effects decreased after discontinuation of therapy, although not to the baseline before the start of treatment (Figure 5B, P1482)**.** In the IFN-gamma ELISPOT assay, specific cells were detectable against PRAME and WT-1, where the number of signals against PRAME was 10 times higher compared to WT-1.

Immune monitoring (including standard immune status, leukemia-specific cell monitoring by ELISPOT and CSA + LAA stimulation) at defined timepoints showed (in contrast to samples collected before treatment and in the patient without Kit M treatment) a continuous increase in DCs.

#### 2.2.4. Patient 1601: Induced Immunological Effects and Transient Reduction in Peripheral Blasts

Shortly before the start of blast-modulatory treatment, the patient received four cycles of Azacitidin (75 mg/m^2^, d1 to d5) with Venetoclax (400 mg/d) added in cycle 5. Treatment was stopped due to progression (nearly 80% blast in PB). As salvage therapy, P1601 received 50 μg/m^2^ GM-CSF, transfused iv. over 4 h for 3 days from day 9 to 11, which was escalated to 75 μg/m^2^ GM-CSF from day 12 to 26. In addition, the patient received PGE_1_ iv. in escalating dose and frequency, as shown in Appendix A. Other than P1482, P1601 received drugs serially within 6 h from day 14 to 26. Stopping rules for the experimental treatment in the case of adverse events were defined as given for patient P1482.

Routine clinical and laboratory parameters showed that the treatment was clinically well tolerated without infusion-related side effects on blood pressure. Neutrophils, hemoglobin, and thrombocyte values stayed low, and the patient had to receive erythrocyte and platelet transfusions. PB blast counts were constantly high (>75%) but decreased soon after the application of Kit M treatment between days 9 and 10 (from 90% to 78%) and between days 17 and 19 (from 95% to 89%). Interestingly, the blast counts increased by the application of prednisolone (applied for a pre-existing COPD) on days 12–14 (Figure 3). On day 29, the patient decided to stop all treatments and died with refractory disease on day 31.

Immune monitoring (including standard immune status and leukemia-specific cell monitoring by ELISPOT and InCyt + LAA stimulation) at defined timepoints showed (other than before treatment and in the patient without Kit M treatment) a slight increase in DCs, proliferating T cells, and stable T_non-naive_ of the CD8 and CD4 lines during Kit M treatment. Th_1_+ and Th_17_+, CD4^+^ T cells and Bcell_memory_ increased during treatment, whereas T_cm_ decreased and T_em_ stayed on a stable line. Stable frequencies of NK cells (either CD161^+^ or CD56^+^), of CIK cells (either CD161^+^ or CD56^+^), and decreased frequencies of iNKT cells (of the NK- as well as the CD3-type) were seen during Kit M treatment (Figure 5A, P1601).

Antigen-specific cells were monitored after LAA stimulation by InCyt. The overall stable frequencies of IFN-γ-producing CD4 and CD8 cells and decreasing frequencies of CIK cells were seen, whereas frequencies of iNKT cells slightly increased during Kit M treatment but decreased under chemotherapy, suggesting a slight in vivo production/activation (potentially leukemia-specific) or at least stable frequencies of cells (Figure 5B, P1601). Specific cells against PRAME and WT-1 could be directly detected only once after treatment by the IFN-gamma ELISPOT assay.

**Figure 3 ijms-25-13469-f003:**
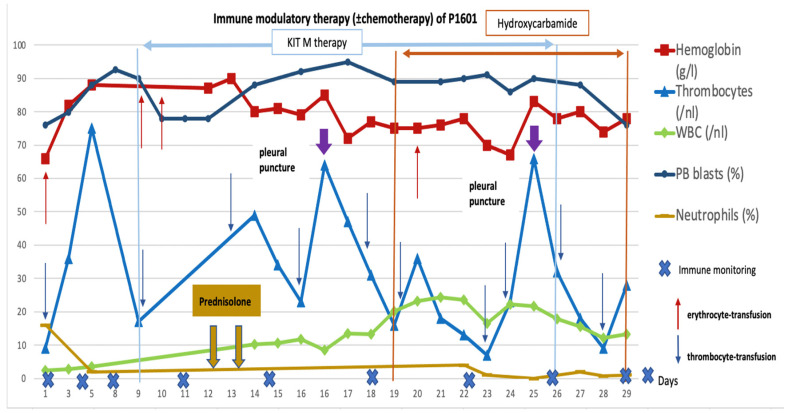
**Clinical course of disease of P1601 during Kit M treatment ± chemotherapy.** No treatment was given between day 1 and 9, Kit M treatment was from day 9 to 26, and chemotherapeutical treatment (hydroxycarbamide) was from day 19 to 29. Patients’ pneumonia was additionally treated daily by prednisolone on day 12–14. Blood cells (thrombocytes, hemoglobin, neutrophils, blasts) in peripheral blood (PB) and leukocytes/white blood cells (WBCs) are given. ↑ Timepoints of erythrocyte transfusions; ↓ timepoints of thrombocyte-transfusions; 
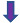
 pleural punctures.

#### 2.2.5. Control Patient 1511: No Immunological or Clinical Response to Treatment

A patient in comparable clinical condition (P1511) did not receive Kit M treatment but conventional chemotherapy instead. He agreed to use his clinical data and blood samples as a control for P1482 and P1601. This patient showed low platelet, hemoglobin, and neutrophil counts, while blast counts persisted in high frequencies throughout chemotherapy (Figure 4).

**Figure 4 ijms-25-13469-f004:**
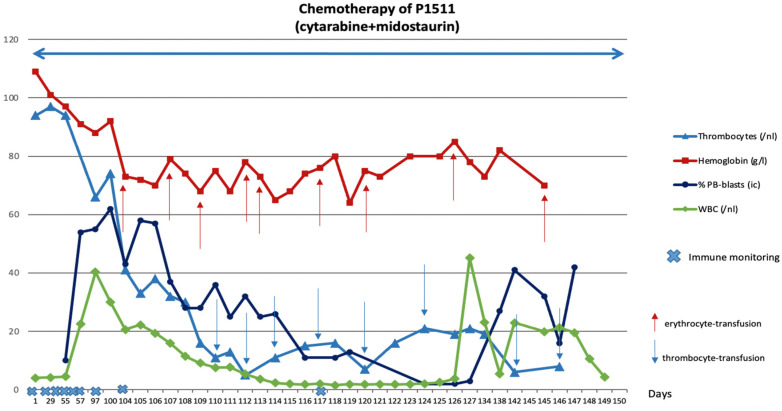
**Clinical course of disease of P1511 during chemotherapy (control without Kit M).** Chemotherapy (cytarabine, midostaurin) was given from the start to the end of observation. Blood cells (thrombocytes, hemoglobin, blasts) in peripheral blood (PB) and leukocytes/white blood cells (WBCs) are given. ↑ Timepoints of erythrocyte transfusion; ↓ timepoints of thrombocyte transfusion.

Immune monitoring (including standard immune status and leukemia-specific cell monitoring by ELISPOT and CSA + LAA stimulation) at defined timepoints showed (other than before treatment and in the patients with Kit M treatment) a decrease in DCs and in proliferating and T_non-naive_ of the CD8 and CD4 lines. Th_1_ and Th_17_ CD4^+^ T cells and T_cm_, T_em_, and Bcell_memory_ decreased in the course of the disease. The frequencies of NK cells decreased, whereas frequencies of iNKT cells were kept stable over 60 days; however, they decreased in the further course of observation (Figure 5A, P1511).

Antigen-specific cells were monitored after LAA stimulation by CSA, varying in general decreasing frequencies of leukemia-specific cells of the T-type, and innate lines were found (Figure 5B, P1511).

**Figure 5 ijms-25-13469-f005:**
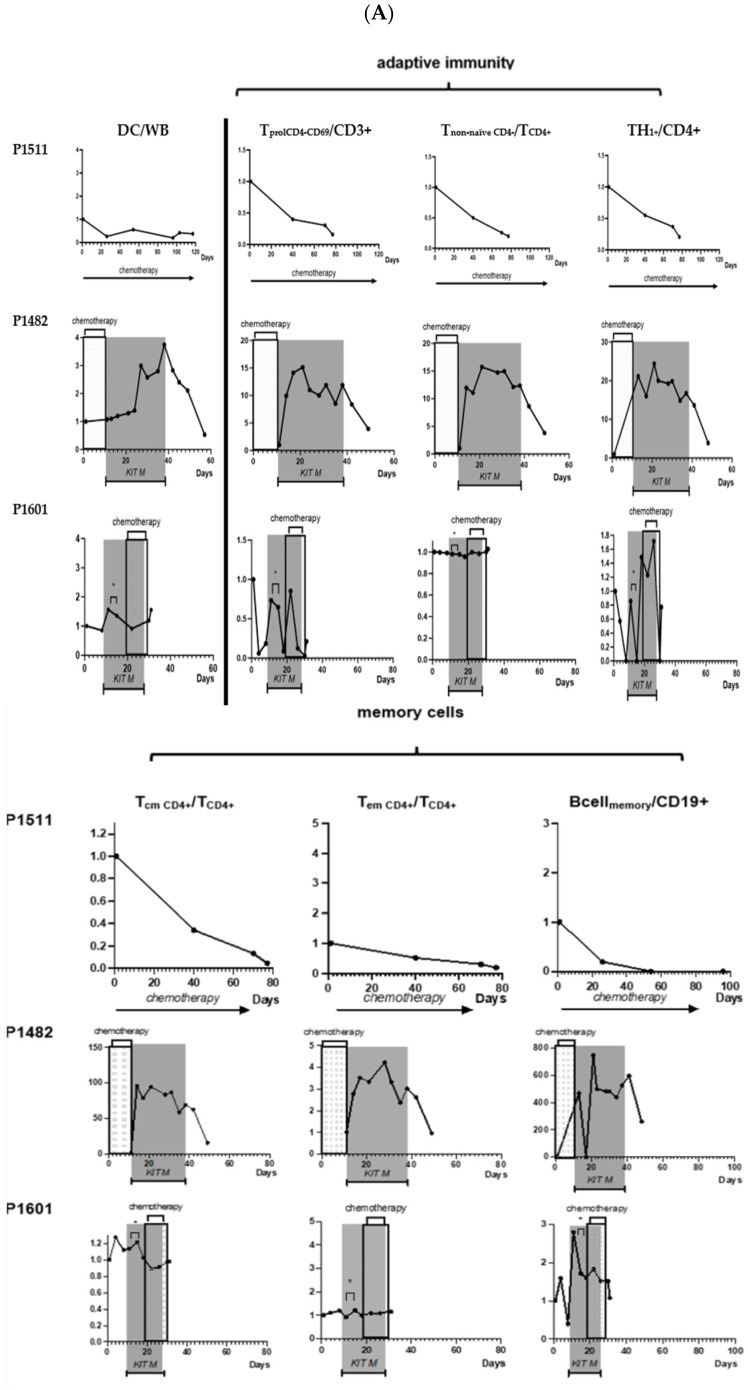
**Immune monitoring under the treatment of therapy refractory patients (P1482, P1601) with Kit M compared to a control patient (P1511).** All values in the course of the disease are given as ‘fold change’ values, referring to the value at the beginning of observation. P1511: chemotherapy during the whole observation time. P1482: low-dose chemotherapy from day 1 to 11, Kit M treatment between day 11 and 38, no treatment from day 38 to the end of observation. P1601: no treatment between day 1 and 9, Kit M treatment from day 9 to 26, chemotherapy from day 19 to 29. * Application of prednisolone from day 12 to 14. Details and abbreviations on cellular subtypes are given in Appendix A. Details on individual treatments are provided in Appendix A. (**A**) Effects of Kit M treatment on patients’ DCs, cells of adaptive immunity, memory cells, and cells of innate immunity in the course of the disease. (**B**) Effects of Kit M treatment (P1482, P1601) (vs. no Kit M treatment (P1511)) on the provision of leukemia-specific cells.

## 3. Discussion

Standard DC-based immunotherapeutic strategies have utilized DC/DC_leu_ production under good manufacturing practice (GMP) conditions, followed by an adoptive transfer to patients [18,22]. However, this procedure is expensive and time-consuming. The direct conversion of (residual) patient blasts in vivo to dendritic cells of leukemic origin, resulting in the activation of (leukemia-specific/antileukemic) immune cells appears to be an elegant solution to simplify this procedure. The idea was based on in vitro experiments, successfully demonstrating the ability of leukemic blasts to convert to functional DC_leu_ [11]. Using PB samples containing leukemic blasts from both diseased animals and patients, we were able to confirm ex vivo previous findings and generate DC/DC_leu_ without inducing blasts’ proliferation using Kit M, a defined combination of the immune response-modifying approved drugs GM-CSF and PGE1. The application of Kit M to blast culture systems led to leukemia-specific/antileukemic immune activation after MLC (enriched with rats’ or patients’ T cells) and especially to improved blast lysis.

Nevertheless, the translation of in vitro findings and postulated mechanisms into an animal model and finally, the clinic remained to be carried out. This prompted us to deduce a Kit M-based treatment strategy for leukemia-diseased rats and therapy refractory AML patients. In this paper, we show first in vivo data on the systemic application of immune response-modifying drugs, given with the intention to generate in vivo DC_leu_-based immune responses in leukemia-diseased rats and, in a second step, end-stage leukemia patients.

### 3.1. Treatment of Leukemia-Diseased Rats with Kit M Was Well Tolerated, Reduced Regulatory T Cells, and Induced Antileukemic Responses

The chosen rat model was qualified to demonstrate the in vivo effects of Kit M. Leukemia-diseased or healthy rats tolerated treatment with single or combined response modifiers very well (no changes in behavior, respiration, weight, skin, etc.). Leukemia-diseased rats showed hematological improvement and immunological effects after treatment with Kit M. Most importantly, as compared to untreated rats, a significant reduction in blasts was seen after two applications of Kit M despite high blast counts in rats’ blood and spleen at the start of treatment. This points to the activation of leukemia-specific effector cells, leading to blast lysis (as seen in rats’ blood and in various human WB models ex vivo [23]).

Furthermore, in contrast to control animals, a reduction in regulatory T cells was seen in the spleen but not in the blood of Kit M-treated rats. This could point to processes leading to reactivated antileukemic responses in vivo, as already shown with specifically induced reactivations against blasts in leukemic patients’ WB samples ex vivo [18]. also found borderline significantly increased frequencies of memory-like T cells after the treatment of rats with Kit M. This might point to an induced immunological memory against blasts in vivo, although definite proof for leukemia specificity is missing due to a lack of a leukemia-specific functional assay. However, based on our ex vivo data on human WB [23], we would suppose that leukemia-specific cells have been induced.

### 3.2. Treatment of Refractory AML Patients with Blast Modulatory Kit M Is Safe, May Induce Platelet Regeneration, and Improve the Composition of (Antileukemic) Immunoreactive Cells

Two AML patients with refractory disease lacking any established treatment options (P1482 and P1601) underwent individualized blast modulatory rescue therapy with Kit M (GM-CSF + PGE_1_). Following extensive discussion with the patient and after obtaining written informed consent, this was the first clinical use of a completely novel treatment concept for AML.

Kit M treatment was well tolerated, without signs of severe adverse events, even in the case of application within only 5 h (in patient 1601). In addition, an increase in mature neutrophile counts (after chemotherapy- and disease-related leukocytopenia) without the induction of blast proliferation was seen in both patients. This might be an effect of GM-CSF and could contribute to minimizing risk for infections [24,25] Moreover, leukemic blasts in PB remained below the (immunological) detection limit over the whole phase of treatment in patient 1482, although 70% BM blasts were detectable before start of Kit M treatment. However, the patient presented with a leukemia relapse one week after the end of Kit M treatment. Since no BM aspiration had been performed during the 4-week treatment, no monitoring of the tumor load (e.g., transient blast decrease) nor immune constellation in bone marrow was possible (Figure 2).

An interesting finding in patient 1601 was the blast reduction between days 9 and 12 under the single treatment with Kit M (in low concentrations). This could be interpreted by an (immunological) antileukemic mechanism induced with a low dosage of Kit M drugs. Between days 12 and 14, prednisolone was given to treat COPD. The immune suppressive effect, however, might have led to a ‘knock out’ of immunological antileukemic effects, leading to an increase in blasts until day 17. Between days 17 and 19, a slight blast reduction was seen, which could be an effect of restored antileukemic immune reaction after the cessation of steroids (Figure 4).

A surprising observation was the rapid recovery of platelets in patient 1482, which made the patient independent of platelet transfusions. This could be a synergistic effect of the combined application of GM-CSF and PGE_1_, since neither GM-CSF nor PGE_1_ are known to stimulate thrombopoiesis. Recently, we have demonstrated that Kit M could have a platelet-stimulating effect on WB and whole bone marrow samples in ex vivo cultures of some, although not all, AML samples in the absence of thrombopoietin agonists/thrombopoiesis-stimulating factors [11]. No recovery of platelets was seen in patient 1601. The requirements for erythrocyte transfusion were not changed in both patients.

Monitoring of the patients’ PB revealed an overall increase in DCs and DC_leu_ in patient 1482, although with variations between samples, which might be explained by DCs’ migration into the tissue. There was no increase in blast proliferation but instead, decreased peripheral blast counts were seen, underlining the safety of the applied drugs. Moreover, our data showed a continuous increase in several cellular subtypes of the adaptive (B- and T-linear) (leukemia-specific) immune system during the treatment phase. This resulted in increased frequencies of B- and T-memory cells, as well as cells of (leukemia-specific) innate immunity. These cells decreased again to a certain degree after the discontinuation of therapy; however, they remained at a higher level compared to the start of treatment. The results suggest an induced or improved antileukemic response of the cellular immune system by Kit treatment, which ultimately may have led to improved clinical and hematological parameters of the patients despite the absence of any further antileukemic treatment. The induction of ‘leukemia-specific’ immune reactions as described here was already described as an important step towards stabilization of the disease or remission (e.g., [26,27,28,29]). It remains to be discussed whether the patient would have had additional benefits from a longer treatment duration or re-challenge with Kit M.

In patient 1601, immunoreactive cells only slightly improved (or at least stable cell counts were seen) in the treatment phase with Kit M (without prednisolone). These results might suggest a (potentially) induced antileukemic response after Kit M application. Accordingly, the activation of (leukemia-specific) cells was only seen in part in this patient, possibly due to high peripheral blast counts (and, consequently, repressed immune cells) during the whole treatment phase.

In contrast to P1482 and P1601, the monitoring of P1511s’ PB revealed low levels of DC and DC_leu_, continuously decreasing frequencies of immune-reactive cells of all lines, namely decreasing proliferating T cells, memory T cells, NK, NKT, and CIK cells, as well as leukemia-specific immune cells. All these findings suggest lower antileukemic competence in this not-kit-treated patient, leading to the immune systems’ inability to cope with the disease (Figure 5).

## 4. Material and Methods

### 4.1. Animal Experiments

#### 4.1.1. Animal Model

Seven-week-old inbred Brown Norway Myeloid Leukemia (BNML) (BN/OrlRj) rats were used as a model for myeloid leukemia that closely resembles the human promyelocytic subtype [21,30]. PVG.1N is a PVG rat strain with the MHC background as in BN rats, in which AML cannot be induced; hence, PVG.1N (RT1^n^) rats were used for safety analyses. The use of animals was approved by the Norwegian Animal Research Authority (NARA) under license numbers 12.4196 and 6060.

BNML cell suspensions were prepared from the spleen of leukemia-diseased rats, washed, and frozen at −80 °C until further use. In male rats, BNML disease was induced by intravenous injection of approximately 8 × 10^6^ BNML cells into the penile vein of anesthetized animals. BNML cells in PB were detectable after 15–17 days. By day 23, a median of 65% blasts were found in PB, and leukemic cells had infiltrated the liver, spleen (88% blasts), and bone marrow (BM, 94% blasts, replacing healthy hematopoiesis). Typical clinical symptoms of rats suffering from leukemia included an enlarged spleen and liver, weight loss, fatigue, and a brittle coat.

#### 4.1.2. Application of Single Response Modifiers to Healthy Rats (Safety Analysis)

To observe the possible side effects of the response modifier, PGE_1_ (c = 1.2 μg) diluted in 500 μL PBS buffer was injected in three anesthetized rats without leukemia. Injections of 500 μL PBS into control rats served as the negative control. Monitoring the rats for possible side effects (weight changes, mobility, interaction with other rats, respiration, skin irritation, and sleep) was performed 2 h, 24 h, and 72 h after injections. Rats were sacrificed using CO_2_ 3 days after the last injection and cell samples were taken to study compositions of hematopoietic cells.

#### 4.1.3. Kit M Treatment of Rats Diseased with Leukemia

To study the possible effects of Kit M on the tumor load as well as on the composition of immunoreactive cells, leukemia-diseased rats were split into two treatment subgroups of three rats each. Fourteen days after the injection of the BNML cells, the rats from group 1 were anesthetized and treated with 1 μg rat GM-CSF and 1.2 μg PGE_1_ (Kit M). Drug combinations were each dissolved in 500 μL PBS before injection. A second dose of drugs was injected four days after the first injection. Animals from group 2 received no drug injections and served as controls.

To monitor the tumor load and immunoreactive cells, rats were sedated by inhalation of Isoflurane Baxter (Baxter International Inc., Deerfield, IL, USA) and 0.2 mL of blood from the lateral tail vein was collected weekly. After 24 days, the rats were sacrificed by asphyxiation with CO_2_. Blood samples were taken by heart punction, and spleens were removed, weighed, and processed for cell preparation shortly after sacrifice. Frequencies of blasts and immune-reactive cells in the blood and spleen were analyzed by flow cytometry.

For an optimal comparison of results, the treatment of all rats was started and rats were sacrificed on the same day.

#### 4.1.4. Flow Cytometric Analyses on Rat Cells

Whole-blood (WB) samples were lysed using RBC lysis buffer, centrifuged, and cells were stained with rat-specific mAbs and incubated for 15 min in the dark on ice. The measurements were performed on fluorescence activated cell sorting (FACS) machines Canto™ (BD, San Jose, CA, USA) or LSR Fortessa™ (BD, San Jose, CA, USA) using the FACS Diva software (Version 6, BD, San Jose, CA, USA). The verification of successful DC/DC_leu_ generation was obtained by staining the cells’ surfaces with the blast marker RM124 (FITC, BD Pharmingen, Franklin Lakes, NJ, USA) and the DC markers CD86 (PE, BD Pharmingen), CD103 (Alexa 647, BD Pharmingen), and MHC class II (Strepatvidin-PerCP, BD Pharmingen) [12,16,21,22,23,24,25,26,27,28,29,30,31]. Moreover, T, NK, and NKT cell subtypes [15,21,31], were quantified, as shown in Appendix A.

### 4.2. Individualized Clinical Treatments in Two Patients with Refractory AML

Two patients with refractory AML without any further treatment options were offered an individualized salvage treatment with Kit M. Treatments had been extensively discussed with the responsible ethical committees and were approved by the patients’ healthcare providers. Before the start of treatment, both patients (P1482, P1601) were elaborately informed by experienced hematologists at several occasions about the experimental nature as well as possible side effects of the treatment, and they provided written informed consent for the treatment as well as examinations on blood samples drawn in addition to routine monitoring. Treatment plans were adapted to individual conditions. For safety reasons, strict stopping rules for the experimental treatment were defined as follows: (1) the discontinuation of GM-CSF therapy in case of reaching values > 10,000 blasts/μL or >50% of leukocytes or in the case of lung toxicities (severe dyspnea, severe affections of blood gas parameters). (2) The discontinuation of PGE_1_ therapy in case of a drop in blood pressure (<100 mmHg systolic or 50 mmHg diastolic) or clinical signs of cardiac failure (dyspnea, pulmonary or peripheral edema). (3) The discontinuation of infusion of both drugs in case of any other toxicities (>CTC2) with restart of infusions only in the case of complete normalization of parameters. A third patient (P1511) in comparable clinical condition, who was treated with palliative therapy only, provided consent for immune monitoring of multiple blood samples drawn in addition to routine blood tests during his clinical course. Repeated blood samples were drawn from all three patients during the course of their respective treatment. Experiments were carried out as approved by the local Ethics Committee (Ludwig Maximilian University, Medical Faculty, Munich, Germany; vote no. 33905).

#### 4.2.1. Flow Cytometrc Analyses on Human Cells

Flow cytometric analyses were used to determine and quantify different phenotypes of leukemic blasts, DC/DC_leu_, T cell subsets, B cells, and monocytes before, during, and after patients’ therapy. Investigated cellular subtypes are described in Appendix A. Panels were labeled with Fluorescein isothiocyanat (FITC), phycoerythrin (PE), tandem Cy7-PE conjugation (Cy7-PE), and allophycocyanin (APC). Examples for gated cell subsets are provided in the Appendix A.

#### 4.2.2. Detection of Antigen-Specific Cells (Interferon Gamma (IFNy)-Cytokine Secretion Assay (CSA), Intracellular IFNy Secretion Assay (InCyt), IFNy-ELISPOT)

To evaluate and quantify IFNy-secreting cells in PB from AML patients during Kit treatment, a CSA and InCyt, as described [23,32], and additionally at some timepoints, IFNy-ELISPOT analyses [33], PMID: 30678050) were performed. WB samples were tested in parallel for IFNγ secretion (with or without prior leukemic antigenic stimulation (LAA, WT1, PRAME)). Detected IFNy-producing cellular subtypes are given in Appendix A.

### 4.3. Statistical Methods

Graphics and statistical analyses were performed with GraphPad Prism 9 (GraphPad Software 68, San Diego). Data acquired by flow cytometry were evaluated with FlowJo (version 7.6.5). Data were presented as mean ± standard deviation of the mean (SEM). Statistical comparisons between experimental groups were performed with the parametric one-way analysis of variance (ANOVA) in combination with a non-parametrical Mann–Whitney test using Microsoft Excel and IBM SPSS Statistics 24 (Release 24.0.0.0, 64-Bit-Version).Differences were considered as ‘not significant’ in cases with *p* values > 0.1, as ‘trend for significant’ (significant *) with *p* values between 0.1 and 0.05, as ‘significant’ (significant **) with *p* values between 0.05 and 0.005, and as ‘highly significant’ (significant ***) with *p* values < 0.005.

## 5. Conclusions

In summary, the present study describes a systematic translation of innovative scientific results from bench to bedside, as well as an example of the development of a new drug based on two approved drugs in an alternative clinical scenario. Within the limitation of limited numbers in both animal experiments and clinical applications, our data suggest that Kit M (containing the clinically approved drugs GM-CSF and PGE1) is able to produce DC/DC_leu_ in vivo in both leukemia-diseased rats and humans. Compared to controls (untreated rats/patient), clinical and hematological antileukemic responses were observed. Kit M as a rescue therapy was shown to be safe, leading to neutrophil and platelet recovery without increasing blast counts. At the same time, immune cells of the adaptive and innate lines were activated and gave rise to memory- and leukemia-specific/antileukemic cells. Based on the data presented here, this in vivo strategy aiming at the conversion of (residual) blasts to leukemia-derived DCs (without the need of GMP procedures) appears to be promising, since DC/DC_leu_ can migrate to tissues and prevent (extramedullary) relapses. DC/DC_leu_ generation is independent of patients’ age, MHC, mutation, cytogenetic risk, transplant, or FAB status (e.g., [11,23] and unpublished data) and could contribute to stabilization of the disease or of remissions by the induction of antileukemic cells and immunological memory.

Nevertheless, we acknowledge the tentativeness of our data, obtained in a small cohort of animals and individuals and meticulously selected patients. Beyond that, as shown in our previous work (e.g., [23]), it seems that the effects of Kit M on the generation of dendritic cells and immunoreactive cells differ slightly between peripheral and bone marrow blood [11], which might explain the finding that the DCleu increase parallels with the clinical response but was finally not able to provide long-term disease control. Larger series are required to confirm our observations and systematically obtain data on immune response modification, clinical safety, and efficacy. Similarly, as observed, clinical and immunological effects were transient, optimized doses and schedules of the applied compounds need to be defined, as well as the combination with other strategies controlling the leukemic proliferation or the application of Kit M in less proliferative stages of the myeloid diseases, e.g., in high-risk myelodysplastic syndrome, hypocellular AML, or a minimal residual disease situation. However, addressing these issues requires a well-designed clinical trial that needs to be based on robust preclinical data. If confirmed in further experiments and clinical studies, the concept may add to the armamentarium of treatments for highly aggressive blood cancer.

## Figures and Tables

**Table 1 ijms-25-13469-t001:** Characteristics of acute myeloid leukemia (AML) patients.

Patient ID	Diagnosis	Stage	Age at Diagnosis (Years)	Sex	ELN Risk Stratification at Diagnosis	First-Line Treatment	Further Treatments	Blast Immune Phenotype (CD-Positivity)	BM/PB Blasts (%) Before Kit M Treatment	Conducted Cell Biological Experiments
P1511	AML	Refractory Relapse	76	m	Intermediate	Decitabine (22 cycles)	Cytarabine + Midostaurin	13, 33, **34**, **117**	54/40	DC, MLC, CTX, CSA
P1482	AML	Refractory Relapse	74	m	Unfavorable	5-Azacytidin (8 cycles)	Second line: Daunorubicin/AraC (2 cycles), followed by HD AraC (three cycles)	13, 15, 33, **34**, 64, **117**	<5/68	DC, MLC, CTX, CSA
							Third line: Decitabine (two cycles)			
							Fourth line: Hydroxyurea/ Cytarabine			
P1601	AML	Refractory Relapse	74	f	Unfavorable	5-Azacytidin (four cycles) and Venetoclax (one cycle)	Hydroxyurea	13, 33, **34**, **117**	90/68	DC, MLC, CTX, InCyt

Abbreviations: AML, acute myeloid leukemia; m, male; f, female; ELN, European Leukemia Net; CD, cluster of differentiation; PB, peripheral blood; BM, bone marrow; DC, dendritic cell culture measurements; MLC, mixed lymphocyte culture; CTX, cytotoxicity assay; CSA, cytokine secretion assay. **Bold** blast markers used to quantify blasts and DC_leu_.

## Data Availability

The original datasets are presented in the article and Appendix A. Further inquiries can be directed to the corresponding author.

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
