# Peer review of "In Vivo Induction of Leukemia-Specific Adaptive and Innate Immune Cells by Treatment of AML-Diseased Rats and Therapy-Refractory AML Patients with Blast Modulating Response Modifiers"

_ijms, 2024, doi:10.3390/ijms252413469_

Round 1

Reviewer 1 Report

Comments and Suggestions for Authors

The manuscript describes a potential new treatment for AML refractory to conventional therapies.

Combined treatment with GM-CSF and PGE1 (Kit M) showed a clear immunomodulatory effect in vitro converting leukemic blast in leukemic DC.

In the manuscript authors explored the in vivo activity of the new treatment.

They show data in a rat model of AML and in 2 treated patients.

Presentation of a limited data set is acceptable in humans but he experiments in the model should be repeated with a more animals and longer follow-up.

Reviewer 2 Report

Comments and Suggestions for Authors

Acute Myeloid Leukemia is a blood disorder in which the hematopoiesis is severely compromised and eventually leads to the expansion of immature blood cells, blasts, of myeloid origin. When compared to other myeloid disorders (e.g., Chronic Myeloid Leukemia) the therapeutic interventions are still poorly successful, thus AML is currently still considered a high-risk disorder. The aberrant genomic landscape underlying AML is heterogeneous. Nowadays the standard therapeutic approach, especially for young patients, forecasts intensive induction chemotherapy followed by hematopoietic stem cell transplantation (HSCT). Nonetheless, unfortunately still too many patients experience relapsed or refractory disease.

In the manuscript titled "In vivo Induction of Leukemia Specific Adaptive and Innate Immune Cells By Treatment of AML-Diseased Rats and Therapy-Refractory AML-Patients with Blast Modulating Response Modifiers" Atzler M. and colleagues report the in-vivo effects of drug-mediated (i.e., GM-CSF and prostaglandins) leukemic dendritic cells induction in AML animal model and patients.

Though, the approach looks promising, elegant, and appealing the findings presented in the current inquiry are still too preliminary and thus poorly convincing.

Based on their previous work titled "Granulocyte-Macrophage-Colony-Stimulating-Factor Combined with Prostaglandin E1 Create Dendritic Cells of Leukemic Origin from AML Patients’ Whole Blood and Whole Bone Marrow That Mediate Antileukemic Processes after Mixed Lymphocyte Culture" it seems that the effects of Kit-M (GM-CSF+prostaglandins) on the generation of dendritic cells and immunoreactive cells differ slightly in between peripheral- and bone marrow-blood. Indeed, though quite similar they are not fully overlapping and, to many extents, this might explain the in-vivo results where the leukemic dendritic cells increase parallels with a decrease in the number of blasts but does not associate with patients' survival.

Honestly, I advise the authors to implement the part concerning dose optimization and schedules. Additionally, the figures in rat models are too small. At least biological triplicates are required. Then, once robust preclinical data are collected, the authors can move to patients. At this point, the number of patients enrolled can be scaled up. Currently, with only two patients drawing significant and robust conclusions is reasonably impossible.

Minor issues concern a few typos scattered throughout the main text and a couple of references not properly formatted.

Comments on the Quality of English Language

The English language is rather fine, however, it would benefit from a kind of "trimming" to be more impactful.
